# Performance Analysis of SiGe-Cladded Silicon MMI Coupler in Presence of Stress

Sneha Kumari [1], Akhilesh Kumar Pathak [2,*], Rahul Kumar Gangwar [3,*] and Sumanta Gupta [1]

1   Department of Electrical Engineering, Indian Institute of Technology, Patna 801106, India
2   International School of Engineering (ISE), Intelligent Control Automation of Process Systems Research Unit, Chulalongkorn University, Bangkok 10330, Thailand
3   Department of Physics and Electronics, Rajdhani College, University of Delhi, New Delhi 110015, India
*   Correspondence: akhileshpathak57@ap.ism.ac.in (A.K.P.); rahul0889@gmail.com (R.K.G.)

**Abstract:** In this study, we demonstrate the influence of operating temperature variation and stress-induced effects on a silicon-on-insulator (SOI)-based multi-mode interference coupler (MMI). Here, SiGe is introduced as the cladding layer to analyze its effect on the optical performance of the MMI coupler. SiGe cladding thickness is varied from 5 nm to 40 nm. Characterization of the MMI coupler for ridge waveguides with both rectangular and trapezoidal sidewall slope angle cross-sections is reviewed in terms of power splitting ratio and birefringence. Stress-induced birefringence as a function of operating temperature and cladding thickness for fundamental mode have been calculated. A trapezoidal waveguide with 40 nm of cladding thickness induces more stress and, therefore, affects birefringence more than a rectangular waveguide of any thickness. Simulation results using the finite element method (FEM) confirmed that operating temperature variation, upper cladding thickness, and its stress effect are significant parameters that drastically modify the performance of an MMI coupler.

**Keywords:** temperature; stress; interference; coupler; finite element method; photonics

## 1. Introduction

Photonic integrated circuits (PIC) faced a stalemate due to high production costs and poor demand until the advent of silicon photonics. This technology has also offered an alternative to high power consumption and the roadblock in large-bandwidth demand in the electronics industry. Silicon photonics is based on a standard CMOS fabrication facility for photonics device fabrication [1–3]. Due to ease of fabrication, many active and passive silicon photonics devices have been designed, such as high-speed and low-power dissipating compact couplers [4], optical filters [1,5–8], and modulators [9,10]. Generally, photonic devices incorporate $SiO_2$ as an upper cladding layer to confine and guide light in the submicron dimension. Moreover, SiGe, due to its nearly matched material properties, bandgap, and lattice constant, has been efficiently integrated into photonic devices. Application of SiGe improves some mediocre properties of silicon, enabling its wide usage in various applications, such as manufacturing of optical sources, optical modulators, and photodetectors. It also hosts optical non-linear properties through bandgap and strain engineering while retaining a cheap and mature fabrication process [11–14]. However, deposition of SiGe over Si owing to a 4.2% lattice mismatch between Si and Ge leads to stress, which needs to be considered while designing any photonic device [5].

In photonic device design flow, there have always been some limitations, challenges, and constraints that are faced by designers that influence the optical performance of a device [15]. Operating temperature fluctuation, geometric variation in waveguide structure, and stress induced by different layers are some of the challenges in silicon photonic device design that can degrade the performance of a device. Temperature sensitivity of

the optical parameter is a crucial issue in integrated photonic devices [16]. This problem is particularly severe in silicon photonics, where high index contrast and the strong temperature-dependent optical characteristics of silicon make photonic devices intensely susceptible to thermal fluctuation. Performance of photonic devices is highly dependent on the properties of the layer constituting the structure. Stress usually originates from lattice mismatch and different coefficients of thermal expansion of different materials used in PIC realization. In recent years, there has been intense research in strained silicon photonics, which has gained considerable interest in a wide field of applications [17–19]. Application of stress has drawn much attention as it influences device functionality by tuning stress in optical devices. Not only undesired effects such as delamination, cracking, and device deterioration but strain can also be used to strengthen desired properties of nanostructured devices [20,21]. In [6], the influence of stress induced by the SiN cladding layer on the SOI Bragg filter has been presented. The impact of stress developed at the core-cladding interface on performance estimation and filter design methodology has been shown [7,8]. In [7], in the presence of stress, the conventional method fails to design a cascaded filter that shows similar performance when analyzed using the finite-element method. In [8], numerical studies show that the estimated changes in resonance wavelength, bandwidth, extinction ratio, and insertion loss for the proposed filter in the presence of stress are 1520 pm, 26%, 36%, and 19%, respectively. In [22], the effect of stress on hybrid plasmonic waveguides having different metal and dielectric types, thicknesses, and geometrical cross-sections has been shown in detail. Here, the stress-induced changes in effective index, propagation loss, propagation length, temperature sensitivity of effective index, and propagation length are 0.35%, 42%, 42%, $0.141 \times 10^{-4}/°C$, and 15.6 nm/°C, respectively. In [23], the role of temperature sensitivity, device geometry, and stress on a $Si_3N_4$-cladded $2 \times 2$ silicon MMI coupler have been presented. Application of stress has also drawn much attention by tuning stress in optical devices, such as birefringence compensation [24,25], thermal expansion tuning of upper cladding film [26], and polarization-independent devices, by eliminating polarization-dependent issues [27,28].

Among many basic structures used in PICs, the MMI coupler has attained much attention regarding a diverse range of applications, including optical sensing, optical interconnect, and optical communications, due to its many-fold advantages, such as large optical bandwidth, low excess loss, simple structure, improved fabrication tolerance, and low polarization dependence [29,30]. However, past studies have shown the impact of stress on single-mode photonic devices. Therefore, it is necessary to demonstrate the effect of cladding-induced stress on functioning of multimode waveguide devices. In this regard, this work has undertaken detailed analysis of cladding-induced stress on the optical performance of an MMI coupler. To show the impact of variation in parameters such as temperature sensitivity, waveguide cross-section geometry, and stress induced by the cladding layer while designing the photonic device, we have introduced these parameters in the design of the MMI coupler and compared them to study the differences in the properties of the traditional and proposed device design. This paper aims to investigate performance characterization of temperature-sensitive MMI couplers using different waveguide cross-section geometry in the presence and absence of stress induced by the upper cladding layer. Section 2 briefly investigates the basic functioning of the MMI coupler. Section 3 discusses the performance of the MMI coupler with and without considering the stress effect imposed by the SiGe cladding layer and analyses the variation in device characteristics. Finally, Section 4 concludes the paper.

## 2. Theory of MMI Couplers

An MMI coupler, owing to its diverse variety of possible power splitting and combining characteristics, has become an important building block in photonic integrated circuits. The working principle of photonic MMI devices is dependent on the self-imaging principle in which the input field is reproduced in single and multiple images at periodic intervals along the propagation direction of the waveguide [29,30]. MMI devices follow the

interference principle of guided modes, where complete constructive interference leads to single or multiple self-image formations. The interference mechanism of the MMI waveguide is highly dependent on the refractive index of core and cladding region, such that a minute change in the refractive index impacts the waveguide's modal interference by a large amount [30]. Conventional MMI coupler consists of the multimode waveguide as the central region, and the single-mode access waveguides are situated at the input and output sides of the multimode waveguide, as shown in Figure 1. The effective MMI coupler width, which is slightly larger than the geometrical MMI coupler width, comes into the picture as the optical mode of the waveguide penetrates into the surrounding material of the rib waveguide and can be written as [29].

$$W_{eff} = Wm - 0.3\,H + \left(\frac{\lambda_0}{\pi}\right)\left(\frac{\eta_c}{\eta_r}\right)^{2\sigma}\left(\eta_r^2 - \eta_c^2\right)^{-0.5} \tag{1}$$

where Wm, H, $\lambda_0$, $\eta_r$, and $\eta_c$ are width of multimode waveguide, device layer thickness, operating wavelength, refractive index of core, and cladding region, respectively. The input optical field is launched by a single-mode input waveguide, and, after entering a multimode waveguide, it excites several modes. The field profile (TE mode) of the multimode waveguide is expressed as [29,30],

$$E_y^m(x,y) = A_m\,\cos\left(\frac{2u_m}{Wm}x - \frac{m\pi}{2}\right) \tag{2}$$

where $u_m = (m+1)\,\pi/2$ represents the mth mode's wavenumber. The overlap integral of the input field and the field of various modes are used to calculate the field amplitudes of the various modes [30],

$$A_m = \frac{2}{W_{eff}}\int_{-W_{eff}/2}^{W_{eff}/2}\psi(x)\,\cos\left[\frac{(m+1)\pi}{Wm}x - \frac{m\pi}{2}\right]dx \tag{3}$$

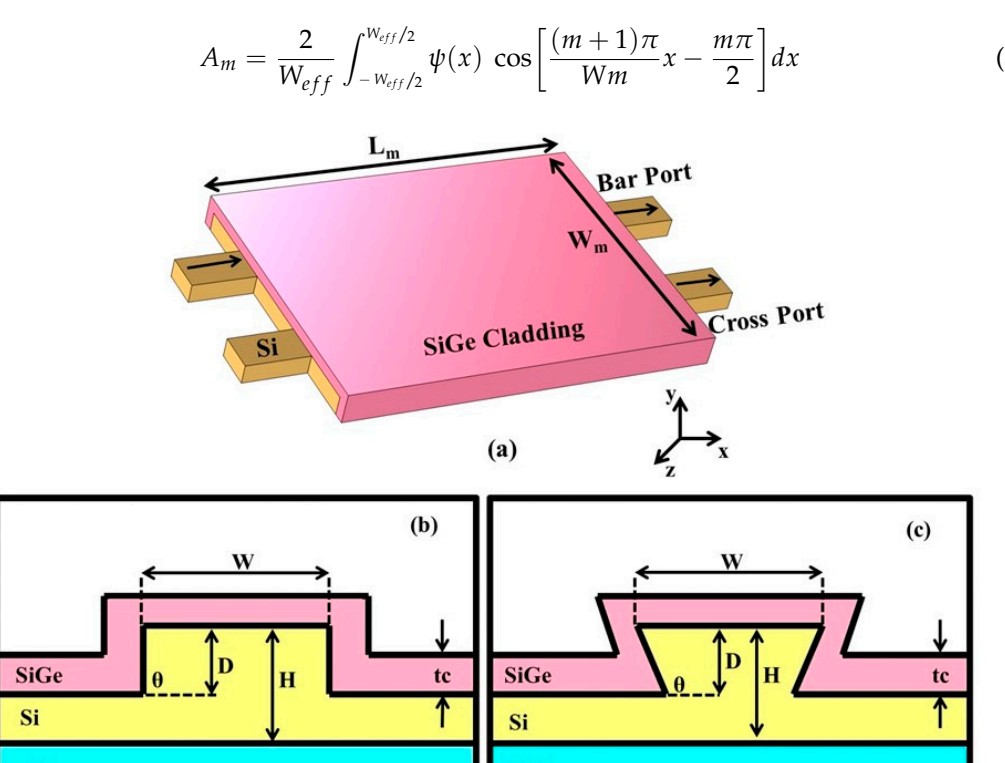

**Figure 1.** Schematic diagram of (**a**) 2 × 2 MMI coupler with SiGe as upper cladding layer with (**b**) rectangular and (**c**) trapezoidal waveguide cross-section.

Here, the input field has a Gaussian profile. The resulting field profile at any specific longitudinal position (z) can be obtained by superimposing all guided modes, which can be represented as

$$
\begin{aligned}
E_{MMI}(x,z) &= \sum_{m=0}^{M} E_y^m(x,y) \\
&= e^{-jk\eta_r z} \sum_{m=0}^{M} A_m \cos\left[\frac{(m+1)\pi}{Wm}x - \frac{m\pi}{2}\right] e^{\left[j\frac{(m+1)^2\pi\lambda}{4\eta_r Wm^2}z\right]}
\end{aligned}
\tag{4}
$$

At the output, which is the length of an integral multiple of $3L_\pi$, different modes after interference regenerate the input field. $L_\pi$ is called beat length and is expressed as [1]

$$
L_\pi = \frac{4\eta_r W_{eff}^2}{3\lambda_0}
\tag{5}
$$

Both waveguides have the same targeted device dimensions, such as device layer thickness H = 220 nm, etch depth D = 170 nm, width of the multimode ridge waveguide $W_m$ = 4 µm, and BOX layer thickness of 1 µm. The input field that has a Gaussian profile with full width at half maximum (FWHM) is 250 nm. BOX layer thickness is the thickness of oxide layer in silicon-on-insulator (SOI) substrate. However, $t_c$ is the thickness of SiGe cladding layer

In this paper, design of a SiGe-cladded silicon 2 × 2 MMI coupler based on the SOI platform is presented. The MMI coupler is designed to offer two power splitting ratios, such as 50:50 and 70:30 splitting ratios at 20 °C. The proposed MMI coupler has a ridge waveguide with two cross-sectional profiles: 90° (rectangular) and 110° (trapezoidal), as shown in Figure 1b,c. Moreover, MMI couplers with both sidewall slope angles have the same device dimensions, waveguide width and device thickness, equal to 4 µm and 220 nm, respectively. MMI couplers are designed to support 16 modes and operate at 1550 nm wavelength. The epitaxial growth of SiGe over Si develops misfit dislocations in the interface between the two and the presence of such misfit dislocations can severely degrade the optical performance of the device. However, the misfit between the epitaxial layers can be made sufficiently small by proliferating the epilayer thickness below a critical value [31]. Moreover, the percentage of Ge concentration in the SiGe-cladded device is found to impact a host of optical properties. The refractive index of $Si_xGe_{1-x}$ increases with the Ge content. As the fraction of Ge increases, the optical mode moves further towards the SiGe cladding layer. Therefore, the Ge fraction should be carefully controlled to achieve complete optical confinement in the core region. In our proposed design, we have taken 0.14 fraction of Ge in the SiGe cladding layer, which made the refractive index of the cladding layer slightly more than the core region.

Finite-element method (FEM)-based simulation was carried out using commercial tool COMSOL Multiphysics to explore the impact of stress caused by the upper cladding layer and operating temperature on the power splitting ratio of the MMI coupler. Real and imaginary refractive index and thermal expansion coefficient (real and imaginary refractive index, density, Young's modulus, Poisson ratio, and thermal expansion coefficient) material properties of Si, SiGe, and $SiO_2$ materials are employed to analyze the proposed MMI coupler in the absence (presence) of stress. Similarly, wave optics/electromagnetic wave, frequency domain (wave optics/electromagnetic wave, frequency domain, and structural mechanics/solid mechanics) physics have been used to examine the proposed device in the absence (presence) of stress. Mode analysis and stationary studies are utilized to extract the modal and stress properties. The mesh density with free-triangular shape used in the simulation is over 5 million for a computation window size of 54 µm × 4 µm, with a maximum element area of around 90 $nm^2$.

### 3. Effect of Si$_{0.86}$Ge$_{0.14}$ Cladding Layer on MMI Coupler in the Absence and Presence of Cladding-Induced Stress

The optical performance of the MMI coupler can be tuned by external parameters, such as temperature, stress, and waveguide cross-section variation. In this section, we are investigating the effect of operating temperature and waveguide cross-section variation with and without incorporating cladding-induced stress. The impact of sidewall slope angles on splitting ratio as a function of operating temperature variation without considering stress for different cladding thicknesses is shown in Figure 2. Due to the limitation of critical thickness, we are restricting the cladding thickness up to 40 nm [5]. For each cladding thickness, the coupler is designed to achieve 50:50 and 70:30 splitting ratios. As the refractive index of the cladding layer is more than the core region, it drags the optical mode towards it from the Si core region. The most significant temperature effect on the MMI coupler is modification of the refractive index as a function of temperature. The thermo-optic effect deals with variation in the refractive index of a material due to temperature variation [32] and is given by

$$n(T) = n_0 + (dn/dt)\Delta T \tag{6}$$

where *n* is the refractive index of the material at room temperature T, (dn/dt) is the thermo-optic coefficient of the given material, and $\Delta$T is the temperature difference between the sample and room temperature.

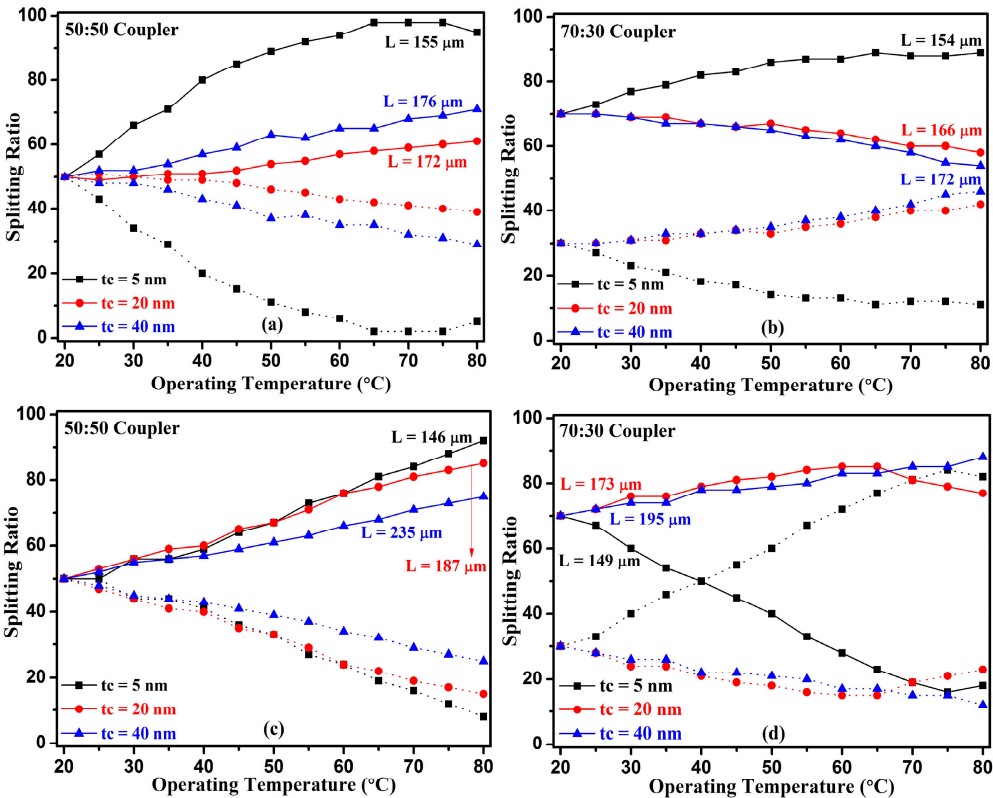

**Figure 2.** Impact of operating temperature on power splitting ratio on rectangular and trapezoidal waveguides. Shown are (**a**) 50:50, (**b**) 70:30 splitting ratio variation for rectangular ($\theta = 90°$) and (**c**) 50:50, (**d**) 70:30 for trapezoidal ($\theta = 110°$) ridge waveguides in the absence of stress. L is the coupling length at which the specific splitting ratio can be acheived for particular SiGe cladding thickness. Power splitting ratios at bar port and cross port are shown by solid and dashed lines, respectively.

A change in the temperature of the waveguide correspondingly changes its refractive index, which, in turn, leads to a change in the interference of the modes and correspondingly changes the self-image formation. The thermo-optic effect of both core and cladding



thickness modifies the effective index of the waveguide. Here, the change in the effective index takes place due to variation in operating temperature from 20 to 80 °C. As the cladding thickness of SiGe continues increasing from 5 to 40 nm, the optical mode extends further in the cladding part; hence, the thermo-optic effect of cladding contributes more. For rectangular and trapezoidal waveguides, SiGe cladding thickness fluctuation from 5 nm to 40 nm would cause a change in $dn_{\text{eff}}/dT$ of fundamental mode from $1.89 \times 10^{-4}/1$ °C to $1.95 \times 10^{-4}/1$ °C, and a temperature fluctuation of 60 °C would cause an effective index change in the order of $1.13 \times 10^{-2}$. From the analysis, it is expected that the change in effective index with respect to temperature variation will increase with a further increase in cladding thickness. For both waveguides, fluctuation of +35 nm in the upper cladding thickness results in effective index change of $1.1 \times 10^{-2}$. For both rectangular and trapezoidal ridge waveguides, the length is optimized at every cladding thickness to couple 50:50 and 70:30 splitting of input light at output ports of the device.

The effects of stress on the identical MMI couplers (as shown in Figure 2) are shown in Figure 3. The figure represents the effect of stress induced by the SiGe cladding layer on both 50:50 and 70:30 power splitting ratios as a function of cladding thickness and operating temperature variation. The stress in the waveguide is mainly determined by the thermal mismatch between the different layers and deposition conditions. The stress-induced refractive index changes are provided by [33,34]

$$n_x - n_0 = -C_1\sigma_x - C_2(\sigma_y + \sigma_z) \tag{7a}$$

$$n_y - n_0 = -C_1\sigma_y - C_2(\sigma_z + \sigma_x) \tag{7b}$$

$$n_z - n_0 = -C_1\sigma_z - C_2(\sigma_x + \sigma_y) \tag{7c}$$

where $\sigma_i$ and $n_i$ (i = x, y, z) represent the stress tensor's component and material's refractive index along the x, y, and z directions, respectively. The $n_0$ shows the refractive index of the material in the absence of stress, whereas $C_1$ and $C_2$ denote the photoelastic constants. However, stress does not only depend on thermal expansion mismatch or lattice mismatch of different layers but is also a function of the waveguide geometry. The epitaxial deposition of a thin SiGe layer grown on a comparatively thick silicon substrate creates intrinsic compressive stress in the cladding layer. Usually, the intrinsic compressive stress of several GPa is ascribed to the 4% lattice mismatch between Silicon and Germanium [5]. In our study, for $Si_{0.86}Ge_{0.14}$ cladding layer, the intrinsic stress of $-1$ GPa has been considered. A change in temperature from deposition to room temperature relaxes compressive stress in the SiGe layer and induces tensile stress in the silicon layer.

For rectangular and trapezoidal waveguide geometry, the film stress ($\sigma_{\text{film}}$) in 5 nm thickness of SiGe film deposited on silicon has a compressive stress of $-806$ (inset in Figure 3a) and $-837$ (inset in Figure 3c) MPa, respectively. In both 90° and 110° sidewall slope angles, a small fluctuation of +35 nm in cladding thickness decreases the film stress in SiGe cladding by approximately 33% (inset in Figure 3b) and 35% (inset in Figure 3d), respectively. Here, varying even a small cladding thickness causes an abrupt change in film stress because the influence of cladding stress becomes large at smaller cladding thickness and the film stress keeps on decreasing with an increase in cladding thickness, which is shown in Figure 3. For rectangular and trapezoidal waveguides, SiGe cladding thickness fluctuation from 5 nm to 40 nm would cause a change in $dn_{\text{eff}}/dT$ of fundamental mode from $1.86 \times 10^{-4}/1$ °C to $1.92 \times 10^{-4}/1$ °C, and a temperature fluctuation of 60 °C would cause effective index change in the order of $1.11 \times 10^{-2}$. For both waveguides, fluctuation of +35 nm in the upper cladding thickness results in effective index change of $1.10 \times 10^{-2}$. Inclusion of stress for both sidewall slope of angle causes a change in effective index of the waveguide mode in the order of approximately $3 \times 10^{-3}$. Here, modification of the effective index is due to inclusion of stress, which varies the optical performance of the device from the unstressed waveguide. Modification of effective index due to both stress and thermal-induced effect varies modal interference, relative phase shift between the propagating modes, and, hence, varies the splitting ratio of input light at output of the

device. When the temperature varies from $T_0$ to $T$, perturbation of the isotropicity of the material index occurs due to mutual thermo-optic and stress-optic effects, which can be defined as [11],

$$n_x = n_0 + B(T - T_0) - C_1\sigma_x - C_2(\sigma_y + \sigma_z) \tag{8a}$$

$$n_y = n_0 + B(T - T_0) - C_1\sigma_y - C_2(\sigma_z + \sigma_x) \tag{8b}$$

$$n_z = n_0 + B(T - T_0) - C_1\sigma_z - C_2(\sigma_x + \sigma_y) \tag{8c}$$

where $n_0$ is the refractive index of the unstressed material at temperature $T_0$ and $B$ is the thermo-optic coefficient of the material.

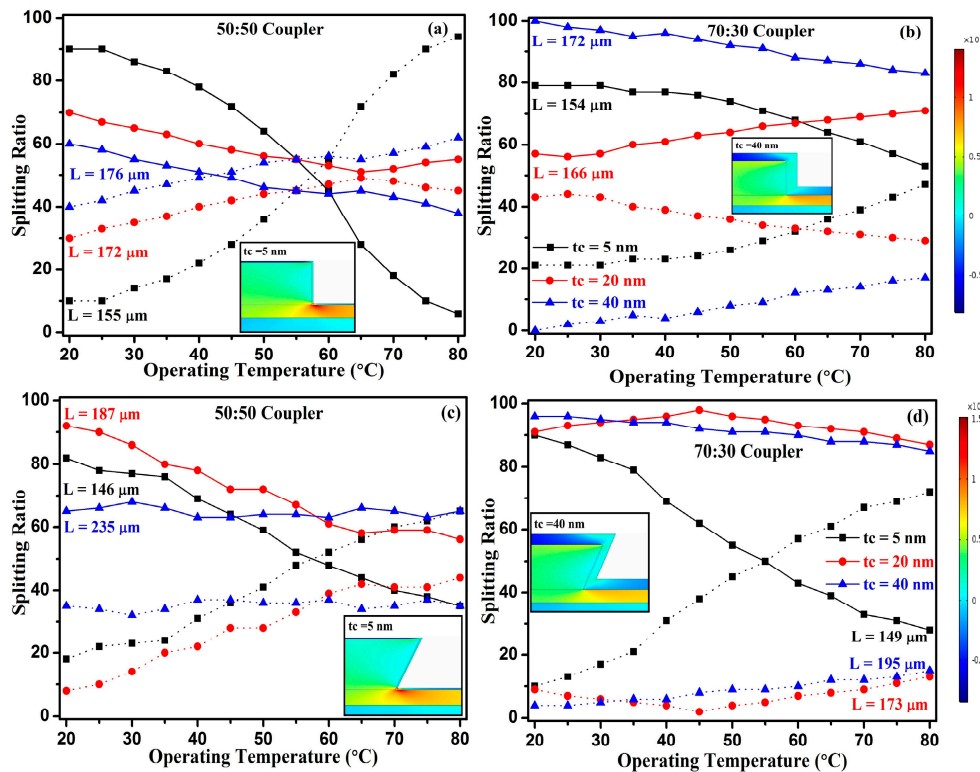

**Figure 3.** Variation in power splitting ratio (**a**) 50:50, (**b**) 70:30 for rectangular (θ = 90°) and (**c**) 50:50, (**d**) 70:30 for trapezoidal (θ = 110°) ridge waveguides as a function of operating temperature in the presence of stress. L is the coupling length at which the specific splitting ratio can be attained for particular SiGe cladding thickness. The solid and dashed lines show power splitting ratios at bar port and cross port, respectively.

The temperature sensitivity of the refractive index affected by both stress-optics and thermo-optic effects can be represented as

$$\frac{dn_x}{dT} = B - C_1\frac{\partial\sigma_x}{\partial T} - C_2\frac{(\partial\sigma_y - \partial\sigma_z)}{\partial T} \tag{9a}$$

$$\frac{dn_y}{dT} = B - C_1\frac{\partial\sigma_y}{\partial T} - C_2\frac{(\partial\sigma_z - \partial\sigma_x)}{\partial T} \tag{9b}$$

Next, we have determined the stress-induced birefringence, which is the difference in ordinary and extraordinary modes as a function of operating temperature and cladding thickness. The stress-induced birefringence calculated using Equation (9a,b) can be written as:

$$\frac{d(n_x - n_y)}{dT} = (C_1 - C_2)\frac{\partial(\sigma_x - \sigma_y)}{\partial T} \tag{10}$$

One of the main causes of thermal stress is temperature change between deposition and operating temperature. Variation in operating temperature from 20 to 80 °C alters the in-plane and out-of-plane stress induced by the upper cladding layer and hence modifies the effective index in the respective direction. These changes, in turn, modify stress-induced birefringence. Cladding thickness plays an essential role in determining birefringence. Birefringence can be effectively modified by altering either cladding thickness ($t_c$) or cladding stress ($\sigma_{film}$). Maximum stress is observed at the corner of the waveguide compared to the center region, which appears in both waveguide cross-section profiles. Therefore, the maximum birefringence variation between both waveguides is found at the corner region. However, at the center of the core region, both waveguides experience similar stress effects, so birefringence is also similar for them, which is evident from Table 1.

**Table 1.** Stress-induced birefringence experienced by the different modes at 20 °C.

| Mode No. | Stress-Induced Birefringence $Si_{0.86}Ge_{0.14}$ (5 nm, θ = 90°) | Stress-Induced Birefringence $Si_{0.86}Ge_{0.14}$ (5 nm, θ = 110°) | Difference in Stress-Induced Birefringence (θ = 110°–90°) |
|---|---|---|---|
| 1 | 0.00828 | 0.00834 | 0.00006 |
| 2 | 0.00769 | 0.00778 | 0.00009 |
| 3 | 0.00708 | 0.00723 | 0.00015 |
| 4 | 0.00666 | 0.00685 | 0.00019 |
| 5 | 0.00625 | 0.00640 | 0.00015 |
| 6 | 0.00592 | 0.00628 | 0.00036 |
| 16 | 0.00275 | 0.00374 | 0.00099 |

For rectangular (θ = 90°) and trapezoidal (θ = 110°) ridge waveguides, cladding thickness fluctuation of +35 nm would cause changes in birefringence of $4.95 \times 10^{-4}$ and $4.75 \times 10^{-4}$, respectively.

Stress-induced birefringence as a function of operating temperature variation from 20 to 80 °C and upper cladding thickness are shown in Figure 4.

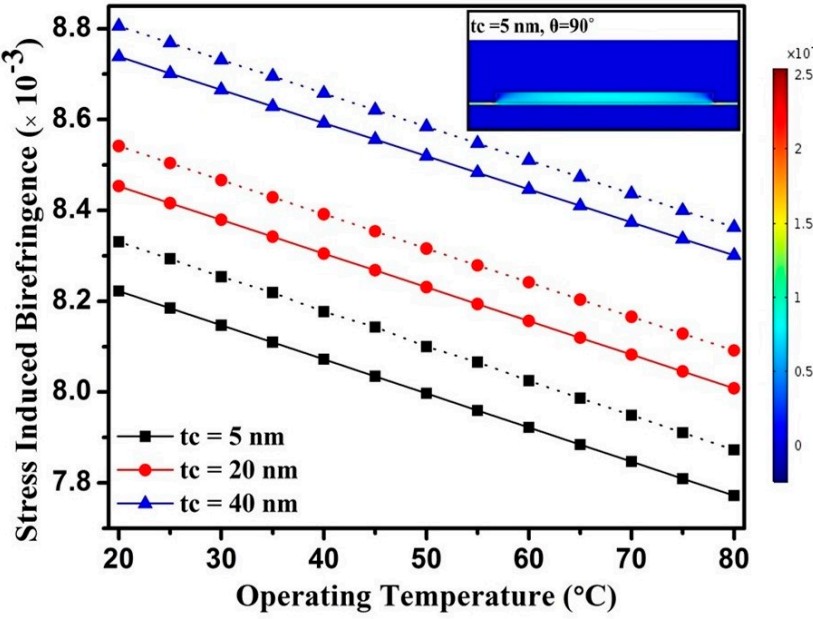

**Figure 4.** Operating temperature influence on birefringence for rectangular (θ = 90°, solid line) and trapezoidal (θ = 110°, dashed line) ridge waveguides for different cladding thicknesses.

The trapezoidal ridge waveguide with a 110° sidewall slope angle induces more stress in the waveguide compared to the ridge waveguide with a 90° sidewall slope angle, so it is expected that a trapezoidal ridge waveguide with 5 nm thickness of cladding layer influences birefringence more than the rectangular ridge waveguide. However, in both the waveguide geometries, further increase in cladding thickness induces almost

the same stress in cladding film and, hence, influences birefringence in a similar manner. Stress-induced birefringence experienced by different modes is shown in Table 1.

Figure 5 illustrates the possible experiment flow to characterize the proposed MMI couplers. The tunable laser source and optical spectrum analyzer can be used to characterize the test structures. The input fiber connects the source via polarization controller (PC). Here, the polarization controller optimizes power coupling. While designing the MMI coupler using the SiGe cladding layer, we must stringently consider the impact of stress and temperature on material properties. Stress imposed by the upper cladding layer at a 110° sidewall slope angle is maximum compared to the 90° sidewall slope angle and hence plays a key role in influencing birefringence. For both waveguide cross-sections, effective thermo-optic constant ($dn_{eff}/dT$) change is similar irrespective of sidewall slope angle, stress, or no stress condition.

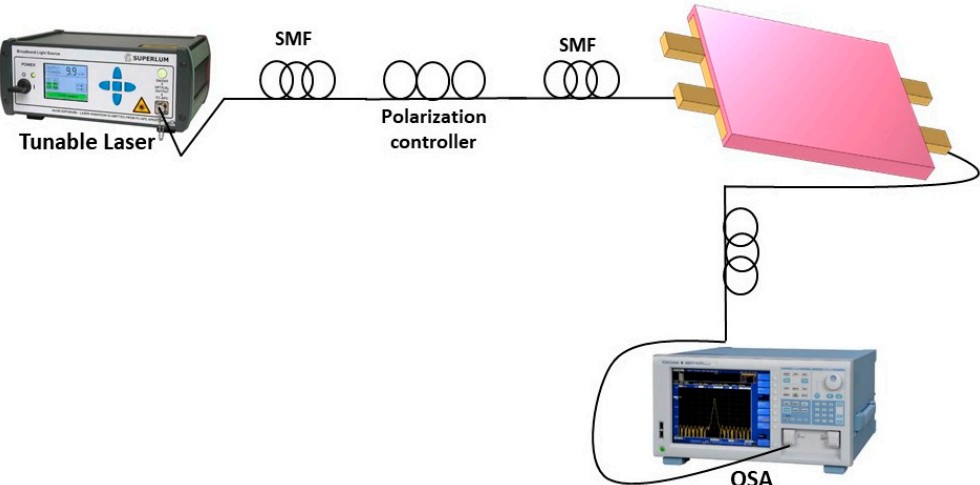

**Figure 5.** Possible experiment layout to characterize the proposed device.

## 4. Conclusions

In this work, an overview of the design challenges encountered by a silicon photonic device designer is demonstrated. We systematically investigated the effects of parameters such as sidewall slope angle, upper cladding material, upper cladding thickness, variation in operating temperature, and stress on the performance of the MMI coupler. Numerical investigation of the influence of stress induced by a SiGe cladding layer on the optical performance of a $2 \times 2$ MMI coupler has been presented in detail. The proposed MMI couplers are designed using two different waveguide cross-sections: rectangular ($\theta = 90°$) and trapezoidal ($\theta = 110°$). To examine the performance of the MMI coupler, two cases were analyzed. The first is for the isotropic refractive index of the material when the stress effects are not considered, while the latter considers stress effects that develop anisotropic refractive index distribution in the material. The temperature sensitivity of the splitting ratio and birefringence are investigated as a function of cladding thickness. For rectangular (trapezoidal) waveguide geometry, the film stress in 5 nm thickness of SiGe film deposited on silicon has a compressive stress of $-806$ MPa ($-837$ MPa), which, with an increase of 35 nm in cladding thickness, decreases the film stress in SiGe cladding by approximately 33% (35%). Hence, the trapezoidal profile can be used for stress-sensing application. Moreover, the trapezoidal waveguide influences birefringence more than the rectangular waveguide of 90° sidewall slope angle. Here, for rectangular and trapezoidal waveguides in the absence (presence) of stress, SiGe cladding thickness fluctuation from 5 nm to 40 nm would cause a change in $dneff/dT$ of fundamental mode from $1.89 \times 10^{-4}/1\,°C$ to $1.95 \times 10^{-4}/1\,°C$ ($1.86 \times 10^{-4}/1\,°C$ to $1.92 \times 10^{-4}/1\,°C$), and a temperature fluctuation of $60\,°C$ would cause an effective index change in the order of $1.13 \times 10^{-2}$ ($1.11 \times 10^{-2}$). The change in effective index with a variation in temperature ($dn_{eff}/dT$) is similar regardless of sidewall slope angle, stress, or no stress condition but is only a function of the thickness of

the upper cladding material. From the findings demonstrated in this paper, we conclude that inclusion of all these parameters in the traditional MMI coupler design leads to large deviation in device characteristics from the pre-defined specification. This work necessitates addressing these factors in device design flow for seamless performance of photonic devices.

**Author Contributions:** Conceptualization, S.K.; Software, S.G.; Validation, A.K.P., R.K.G. and S.G.; Formal analysis, S.K.; Investigation, S.K.; Resources, S.G.; Writing—original draft, S.K.; Writing—review & editing, A.K.P. and R.K.G.; Supervision, S.G.; Funding acquisition, A.K.P. and R.K.G. All authors have read and agreed to the published version of the manuscript.

**Funding:** This research received no external funding.

**Data Availability Statement:** Not Applicable.

**Conflicts of Interest:** The authors declare no conflict of interest.

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
