# Peer review of "Performance Analysis of SiGe-Cladded Silicon MMI Coupler in Presence of Stress"

_computation, doi:10.3390/computation11020034_

Round 1
Reviewer 1 Report
Dear Editor,
In this study, authors demonstrated the influence of operating temperature variation and stress-induced effects on the SOI based MMI. In addition, SiGe is introduced as the cladding layer to analyze its effect on the optical performance of the MMI coupler. This is a simulation work. I have some comments as follows.
a. Authors should give comparisons with different numerical methods.
b. They may have a comprehensive conclusion.
Author Response
All the authors are highly thankful to the respected Editor and Reviewers for their kind and valuable suggestions on the manuscript which helps to improve the article and make it more reader-friendly. We thank the reviewer for reading our paper carefully and giving positive comments. We also appreciate your clear and detailed feedback and hope that the explanation has fully addressed all of your concerns. In the remainder of this letter, we discuss each of your comments individually along with our corresponding responses. The corrections and changes are highlighted in yellow in the revised manuscript.
Reviewer#1
In this study, authors demonstrated the influence of operating temperature variation and stress-induced effects on the SOI based MMI. In addition, SiGe is introduced as the cladding layer to analyze its effect on the optical performance of the MMI coupler. This is a simulation work. I have some comments as follows:
Q1. Authors should give comparisons with different numerical methods.
Ans: In recent years, there has been intense research in strained silicon photonics which has gained considerable interest in a wide field of applications [17-19]. The application of stress has drawn a lot of attraction as it influences the device functionality by tuning stress in optical devices. The proposed work shows the effect of stress on SiGe cladded MMI coupler. For rectangular (trapezoidal) waveguide geometry, the film stress in 5 nm thickness of SiGe film deposited on silicon has a compressive stress of -806 MPa (-837 MPa), respectively which with increase of 35 nm in cladding thickness decreases the film stress in SiGe cladding by approximately 33% (35%). Here, for rectangular and trapezoidal waveguides in the absence (presence) of stress, SiGe cladding thickness fluctuation from 5 nm to 40 nm would cause the change in dneff/dT of fundamental mode from 1.89 × 10-4/ 1°C to 1.95 × 10-4/ 1°C (1.86 × 10-4/ 1°C to 1.92 × 10-4/ 1°C) and a temperature fluctuation of 60°C would cause the effective index change in the order of 1.13 × 10-2 (1.11 × 10-2). The findings of this paper inform, the trapezoidal waveguide cross-section of 110° sidewall slope angle induces more stress in the cladding film and hence photonic device with a trapezoidal profile can be used for stress-sensing application. Also, a trapezoidal waveguide influences birefringence more than a rectangular waveguide of 90° sidewall slope angle.
Similarly, in [1], a numerical study of SiO2 cladded Si MMI coupler has been presented. The FEM-based simulation revealed that the power splitting ratio of the MMI coupler changes drastically (as high as 50% for 50:50 coupler and 25% for 70:30 coupler) in the presence of cladding induced stress. In presence of cladding induced stress for rectangular (trapezoidal) waveguide cross-section, the neff value for fundamental mode changes from 2.84634 (2.84655) to 2.84658 (2.84679) when cladding thickness changes from 400 nm to 800 nm.
In [7], the effect of stress on cascaded filters each having periodic Si3N4 cladding on a silicon strip waveguide has been reported. The study reveals that the filter, which is designed using conventional TMM for a set of target specifications, shows the similar optical performance when estimated using finite-element method (FEM) in absence of stress. In presence of stress, the conventional TMM method fails to design the cascaded IBG filter that shows the dissimilar performance when analysed using FEM.
In [8], study of stress effect on optical performance of surface‑corrugated hybrid plasmonic IBG filter has been demonstrated. In this study, two cross-sectional geometries of HPW namely rectangular and triangular are considered to investigate its effect on proposed filters. For all designed filters, the triangular waveguide induces the highest magnitude of stress as compared to the triangular waveguide. The numerical studies show the estimated changes in resonance wavelength, bandwidth, extinction ratio and insertion loss for the proposed filter in presence of stress are 1520 pm, 26%, 36%, and 19%, respectively.
In [22], the impact of stress on the optical performance of a hybrid plasmonic waveguide has been shown. Here, the optical performances of two HPWs having rectangular and triangular cross-section have been investigated using the finite-element method. The largest magnitude of stress is exerted by the triangular waveguide and the stress-induced changes in the effective index, propagation loss, propagation length and temperature sensitivity of effective index, and propagation length are 0.35%, 42%, 42%, 0.141 × 10-4/â—¦C and 15.6 nm/â—¦C respectively.
The numerical computation of temperature sensitivity of silicon 2×2 multi-mode interference (MMI) coupler having silicon nitride as an upper cladding layer has been presented in [23]. Rectangular and trapezoidal waveguides with upper cladding thickness varied from 300 nm to 800 nm is employed. The rectangular (trapezoidal) sidewall slope angle-based ridge waveguide with the cladding thickness of 300 nm induces 1008 GPa (1020 GPa) stress and varies splitting ratio and birefringence more in trapezoidal waveguide configuration as compared to the rectangular based MMI coupler.
The fact has been included in the introduction section of revised manuscript.
Q2. They may have a comprehensive conclusion.
Ans: Authors are highly thankful to reviewer's suggestion. A comprehensive conclusion has been provided in the revised manuscript.
Reference:
[1] S. Kumari and S. Gupta, “Cladding stress induced performance variation of silicon mmi coupler,” Photonics and Nanostructures-Fundamentals and Applications, vol. 33, pp. 55–65, 2019.
[7]. S. Kumari, S. Gupta, Design of narrow bandwidth Si3N4 stressor cladded cascaded IBG filter, Optik 254 (2022), 168564
[8]. S. Kumari, S. Gupta, "Study of Stress Effect on Optical Performance of Surface-Corrugated Hybrid Plasmonic IBG Filter", Plasmonics 17, 2022, pp.339–348.
[17]. C. Schriever, C. Bohley, J. Schilling, and R. B. Wehrspohn, “Strained silicon photonics,” Materials, vol. 5, no. 12, pp. 889– 908, 2012.
[18]. R. S. Jacobsen, K. N. Andersen, P. I. Borel, J. Fage-Pedersen, L. H. Frandsen, O. Hansen, M. Kristensen, A. V. Lavrinenko, G. Moulin, H. Ou, C. Peucheret, B. Zsigri, and A. Bjarklev, “Strained silicon as a new electro-optic material,” Nature, vol. 441, no. 7090, pp. 199–202, 2006.
[19]. M. Cazzanelli, F. Bianco, E. Borga, G. Pucker, M. Ghulinyan, E. Degoli, E. Luppi, V. Véniard, S. Ossicini, D. Modotto, S. Wabnitz, R. Pierobon, and L. Pavesi, “Second-harmonic generation in silicon waveguides strained by silicon nitride,” Nat. Mater., vol. 11, no. 2, pp. 148–154, 2011.
[22]. S. Kumari, S. Gupta, "Performance Estimation of Hybrid Plasmonic Waveguide in Presence of Stress", Plasmonics 16, pp. 359–370, 2021.
[23]. S. Kumari, S. Gupta, Study on temperature sensitivity of Si3N4 cladded silicon 2 × 2 MMI coupler, 4th International Conference on Opto-Electronics and Applied Optics (Optronix) (2017) 1–5.

Reviewer 2 Report
The paper deals with the influence of temperature and stress on the splitting ratio of the MMI coupler with two different profiles. All the results were obtained by FEM-based simulation using COMSOL Multiphysics tool. Despite the quite well organization of the paper and easy following of the authors‘ approach, I found several areas that could be improved.
I recommend describing the meaning of the Wm, H as well as the other symbols in Eq.(1).
In Fig.1 are W and Wm the same quantities?
I would expect some Reference to the Eq. (5).
Line 123 – what does „BOX layer thickness“ mean? Is it the same as „tc“ in Fig.1b,c?
Lines 142 and 171 – How can the refractive index of the cladding layer be higher than that of the core? It does not correspond to waveguiding effect.
Please, check the grammar, mainly at lines: 89,90, 95, 126,150-152, 183, 212, 243, 254, 271-272, 299-303, 317-319.
Author Response
All the authors are highly thankful to the respected Editor and Reviewers for their kind and valuable suggestions on the manuscript which helps to improve the article and make it more reader-friendly. We thank the reviewer for reading our paper carefully and giving positive comments. We also appreciate your clear and detailed feedback and hope that the explanation has fully addressed all of your concerns. In the remainder of this letter, we discuss each of your comments individually along with our corresponding responses. The corrections and changes are highlighted in yellow in the revised manuscript.
Reviewer#2
The paper deals with the influence of temperature and stress on the splitting ratio of the MMI coupler with two different profiles. All the results were obtained by FEM-based simulation using COMSOL Multiphysics tool. Despite the quite well organization of the paper and easy following of the authors ‘approach, I found several areas that could be improved:
Q1. I recommend describing the meaning of the Wm, H as well as the other symbols in Eq.(1).
Ans: Authors are thankful to reviewer’s suggestion; we have included these notations in the revised manuscript.
Q2. In Fig.1 are W and Wm the same quantities?
Ans: W and Wm are the same quantities in Fig. 1 and the typo has been corrected in the revised manuscript.
Q3. I would expect some Reference to the Eq. (5).
Ans: The reference has been provided in the revised manuscript.
Q4. Line 123 – what does „BOX layer thickness“ mean? Is it the same as „tc“ in Fig.1b,c?
Ans: BOX layer thickness is the thickness of buried oxide layer in silicon-on-insulator (SOI) substrate. However, tc is the thickness of SiGe cladding layer. The line has been modified in the theory of MMI coupler section in the revised manuscript.
Q5. Lines 142 and 171 – How can the refractive index of the cladding layer be higher than that of the core? It does not correspond to waveguiding effect.
Ans: The last decade has seen immense interest of integrating SiGe material on silicon photonic platform. The application of SiGe on Si has been the key enabler of various active devices. It improves some mediocre properties of silicon which allows its wide usage in various applications such as the manufacturing of optical sources, optical modulators, and photodetectors [1,2,3,4,5]. The SiGe fractional value decide the refractive index and hence the optical mode confinement in the cladding layer. Therefore, the particular value of SiGe is decided by its intended application [1,2,3,4,5]. The higher value of Ge in SiGe cladding attracts most of the mode confienment in cladding region and hence can be used as sensor, whereas, the smaller or middle value of SiGe are used to design active devices on silicon photonic platform [2,3,4,5]. In the proposed work, we have taken 0.14 fraction of Ge in the SiGe cladding layer which made the refractive index of the cladding layer slightly more than the core region. This work aims to show the effect of stress induced by SiGe cladding layer on silicon platform, which deviates the optical functioning of the device from the intended function. The fact along with reference has been added in the revised manuscript.
Q6. Please, check the grammar, mainly at lines: 89,90, 95, 126,150-152, 183, 212, 243, 254, 271-272, 299-303, 317-319.
Ans: Auhtors are thankful to reviewer’s suggestions. We have revised the grammatical error in the manuscript.
Referecnce:
[1] Zhiping Zhou, Bing Yin and Jurgen Michel, On chip light sources for silicon photonics, Light: Science & Applications, Nov. 2015, 4, e358; doi:10.1038/lsa.2015.131.
[2] S. Cho, J. Park, H. Kim, R. Sinclair, B.-G. Park, and J. S. Harris Jr., “Effects of germanium incorporation on optical performances of silicon germanium passive devices for group-IV photonic integrated circuits,” Photonics and Nanostructures: Fundamentals and Applications, Vol. 12, No. 1, pp. 54–68, Feb. 2014.
[3] K. Hammani, M. A. Ettabib, A. Bogris, A. Kapsalis, D. Syvridis, M. Brun, P. Labeye, S. Nicoletti, D. J. Richardson, and P. Petropoulos, “Optical properties of silicon germanium waveguides at telecommunication wavelengths,” Opt. Express 21(14), pp. 16690–16701, 2013.
[4] S. Assali, S. Koelling, Z. Abboud, J. Nicolas, A. Attiaoui, O. Moutanabbir. (2022) 500-period epitaxial Ge/Si0.18Ge0.82 multi-quantum wells on silicon. Journal of Applied Physics 132:17, 175304, 2022.
[5] Pandraud, G.; Milosavljevic, S.; Sammak, A.; Cherchi, M.; Jovic, A.; Sarro, P. Integrated SiGe Detectors for Si Photonic Sensor Platforms. Proceedings, 1, 559, 2017. https://doi.org/10.3390/proceedings1040559

Reviewer 3 Report
In this paper, the influence of temperature and stress on MMI coupler operation are investigated by numerical simulation with the commercial software COMSOL multiphysics. The necessity of considering stress effect is insisted. However, I cannot find enough novelty for publishing this paper. Although utilizing the sensitivity to stress, stress-sensing application is suggested, how it works should be presented in the actual application. I think that the accuracy of this analysis is also not confirmed in this paper. The simulation condition for stress analysis has to be provided in detail.
Author Response
All the authors are highly thankful to the respected Editor and Reviewers for their kind and valuable suggestions on the manuscript which helps to improve the article and make it more reader-friendly. We thank the reviewer for reading our paper carefully and giving positive comments. We also appreciate your clear and detailed feedback and hope that the explanation has fully addressed all of your concerns. In the remainder of this letter, we discuss each of your comments individually along with our corresponding responses. The corrections and changes are highlighted in yellow in the revised manuscript.
Reviewer#3
In this paper, the influence of temperature and stress on MMI coupler operation are investigated by numerical simulation with the commercial software COMSOL multiphysics. The necessity of considering stress effect is insisted. However, I cannot find enough novelty for publishing this paper.
Ans: All the authors are highly thankful to reviewer’s suggestion over the novelty. We would like to bring the attention of reviewer’s towards the key novelty of the work which is mentioned below for reviewer’s consideration.
The last decade has seen immense interest of integrating SiGe material on silicon photonic platform to enable various active devices such as optical sources, optical modulators, and photodetectors [1,2,3,4,5]. However, the deposition of SiGe over Si owing to a 4.2% lattice mismatch between Si and Ge leads to stress which needs to be considered while designing any photonic device. However, the investigation which shows the effect of this induced stress on the device performance is missing in the literatures. In this regard, the novelty of this work is that it shows the effect of SiGe induced stress on the optical functioning of the photonic device. From the findings, we have observed a substantial change in the performance of MMI coupler in the absence and presence of stress. The proposed study shows the necessity of considering the stress effects while designing and estimating of photonic devices having different materials in their structure. The MMI coupler having trapezoidal waveguide configuration shows larger stress and therefore can be used in the stress-sensing application. However, the stress sensing is the different domain of research which we will surely consider in our future publication, as this work shows the impact of stress on device performance.
Q1. Although utilizing the sensitivity to stress, stress-sensing application is suggested, how it works should be presented in the actual application.
Ans: All the authors are thankful to reviewer’s suggestion and we really welcome the idea of the stress sensing investigation. We would like to mention that this work shows the impact of stress on device optical performance. However, I would like to declare that the co-authors have expertise in photonics based sensors and the sensing domain. As this is a preliminary work aiming to see the device performance, therefore we will surely implement the suggestion in our on-going work to show its application in stress sensing.
Q2: I think that the accuracy of this analysis is also not confirmed in this paper.
Ans: All the authors are thankful for reviewer’s suggestion, we would like to clarify that the computation works have been repeated multiple times to confirm the accuracy. There’s no significant variation in the result was observed throughout the repetition.
Q3: The simulation condition for stress analysis has to be provided in detail.
Ans: All the authors are highly thankful to reviewer’s suggestion. The simulation conditions are shown below for the reviewer’s consideration and the same has been included in the revised manuscript:
“The finite-element method (FEM) based simulation was carried out using the commercial tool COMSOL Multiphysics to explore the impact of stress caused by the upper cladding layer and operating temperature on the power splitting ratio of the MMI coupler. Real and imaginary refractive index and thermal expansion coefficient (real and imaginary refractive index, density, Young’s modulus, Poisson ratio and thermal expansion coefficient) material properties of Si, SiGe and SiO2 materials are employed to analyze the proposed MMI coupler in the absence (presence) of stress. Similarly, wave optics-electromagnetic wave, frequency domain (wave optics-electromagnetic wave, frequency domain and structural mechanics-solid mechanics) physics have been used to examine the proposed device in the absence (presence) of stress. Mode analysis and stationary studies are utilized to extract the modal and stress properties. The mesh density with free-triangular shape used in the simulation is over 5 million for a computation window size of 54 µm × 4 µm, with a maximum element area of around 90 nm2.”
References:
[1] Zhiping Zhou, Bing Yin and Jurgen Michel, On chip light sources for silicon photonics, Light: Science & Applications, Nov. 2015, 4, e358; doi:10.1038/lsa.2015.131.
[2] S. Cho, J. Park, H. Kim, R. Sinclair, B.-G. Park, and J. S. Harris Jr., “Effects of germanium incorporation on optical performances of silicon germanium passive devices for group-IV photonic integrated circuits,” Photonics and Nanostructures: Fundamentals and Applications, Vol. 12, No. 1, pp. 54–68, Feb. 2014.
[3] K. Hammani, M. A. Ettabib, A. Bogris, A. Kapsalis, D. Syvridis, M. Brun, P. Labeye, S. Nicoletti, D. J. Richardson, and P. Petropoulos, “Optical properties of silicon germanium waveguides at telecommunication wavelengths,” Opt. Express 21(14), pp. 16690–16701, 2013.
[4] S. Assali, S. Koelling, Z. Abboud, J. Nicolas, A. Attiaoui, O. Moutanabbir. (2022) 500-period epitaxial Ge/Si0.18Ge0.82 multi-quantum wells on silicon. Journal of Applied Physics 132:17, 175304, 2022.
[5] Pandraud, G.; Milosavljevic, S.; Sammak, A.; Cherchi, M.; Jovic, A.; Sarro, P. Integrated SiGe Detectors for Si Photonic Sensor Platforms. Proceedings, 1, 559, 2017. https://doi.org/10.3390/proceedings1040559
